# Identification of Novel Natural Inhibitors to Human 3-Phosphoglycerate Dehydrogenase (PHGDH) for Cancer Treatment

**DOI:** 10.3390/molecules27186108

**Published:** 2022-09-19

**Authors:** Ayesha Sadiqa, Azhar Rasul, Mudassir Hassan, Salma Sultana, Farhat Jabeen

**Affiliations:** Department of Zoology, Faculty of Life Sciences, Government College University Faisalabad, Faisalabad 38000, Pakistan

**Keywords:** tumor metabolism, PHGDH, phytochemicals, molecular docking, in silico analyses

## Abstract

Targeting the serine biosynthesis pathway enzymes has turned up as a novel strategy for anti-cancer therapeutics. 3- Phosphoglycerate dehydrogenase (PHGDH) is the rate-limiting enzyme that catalyzes the conversion of 3-Phosphoglyceric acid (3-PG) into 3-Phosphohydroxy pyruvate (3-PPyr) in the first step of serine synthesis pathway and perform a critical role in cancer progression. PHGDH has been reported to be overexpressed in different types of cancers and emerged as a novel target for cancer therapeutics. During this study, virtual screening tools were used for the identification of inhibitors of PHGDH. A library of phenolic compounds was docked against two binding sites of PHGDH using Molegro Virtual Docker (MVD) software. Out of 169 virtually tested compounds, Salvianolic acid C and Schizotenuin F possess good binding potential to co-factor binding site of PHGDH while Salvianolic acid I and Chicoric acid were identified as the best binding compounds toward the substrate binding site of PHGDH. The top selected compounds were evaluated for different physiochemical and ADMET properties, the obtained results showed that none of these hit compounds violated the Pfizer Rule and they possess acceptable ADMET profiles. Further, a commercially available hit compound, Chicoric acid, was evaluated for its anti-cancer potential against PHGDH-expressing gastric cancer cell lines (MGC-803 and SGC-7901) as well as cell lines with low expression of PHGDH (MCF-7 and MDA-MB2-31), which demonstrated that Chicoric acid possesses selective cytotoxicity toward PHGDH expressing cancer cell lines. Thus, this study has unveiled the potential of phenolic compounds, which could serve as novel candidates for the development of PHGDH inhibitors as anti-cancer agents.

## 1. Introduction

Cancer is a multifactorial and highly proliferative disease that is characterized by an uncontrolled division of cells. It is the leading source of death around the globe [1]. During 2020, approximately 9.9 million cancer-related deaths and 19.3 million newly diagnosed cases were recorded worldwide. It is estimated that 58.3% of cancer deaths occurred in Asia in 2020 and this burden is expected to increase in the coming years [2]. Currently, cancer is treated by different strategies which include surgery, radiotherapy, chemotherapy, and immunotherapy. Intra-tumoral heterogeneity and emerging tumor resistance toward chemo-drugs are major hurdles to successful cancer treatment [3,4]. Therefore, there is an urgent need for the development of new therapies for the treatment of cancer patients. 

Metabolic reprogramming is the most crucial feature as well as an emerging hallmark of cancer [5]. To satisfy the demands of highly proliferating cells, cancer cells reprogram their metabolic pathways [6]. Tumor cell produce energy in a very unique manner by enhancing glucose uptake along with the consumption of amino acid, which includes glutamines, however, high consumption of glutamine or glucose is not sufficient to support the cancer progression. It has been reported that the survival of tumors also depends on the non-glutamine amino acids such as serine which provides a bulk amount of carbon and nitrogen units to meet the biosynthetic needs of cancer cells [7]. At the cellular level, non-essential amino acids like serine can either be imported from the medium or synthesized from the glycolytic intermediate 3-Phosphoglyceric acid (3-PG) [5]. Serine is the central node for biosynthesis and its metabolism is frequently deregulated in cancers [8]. 3-Phosphoglycerate is a glycolytic intermediate product and it is the precursor of serine biosynthesis via three consecutive enzymatic reactions branching off the glycolytic pathway. In the first step of this reaction, cells use NAD+-dependent enzyme 3-Phosphoglycerate dehydrogenase (PHGDH) and it catalyzes the 3-Phosphoglyceric acid into the 3-Phosphohydroxy Pyruvate. PHGDH is the first rate-limiting enzyme for the entry of 3-PG into the Serine synthesis pathway (SSP) [9,10,11]. The overexpression of PHGDH has been reported in different cancer cell lines such as lung cancer, glioma, gastric cancer breast cancer, and colon cancer [12]. Knockdown of PHGDH in an in-vivo mouse model played a significant role in the reduction of cancer proliferation and prolonged the survival of the tumor-bearing animal [13] suggesting that PHGDH inhibition have the potential to halt the growth of tumors. 

Our present study investigated the PHGDH binding potential of phenolic compounds library through computer-aided molecular docking. After a screening of the phenolic compounds library, six top-ranking compounds were selected. Further, the ADMET and physiochemical properties of hits were also investigated. MTT assay analysis of a commercially available compound, Chicoric acid, against gastric cancer cells identified Chicoric acid as a potent anti-cancer agent for the treatment of cancers with overexpression of PHGDH. This study has identified natural inhibitors of PHGDH for the development of safer and selective cancer therapeutics. 

## 2. Results

### 2.1. In-Silico-Based Screening of Phenolic Compounds Library against Coenzyme and Substrate Binding Sites of Human 3-Phosphoglycerate Dehydrogenase (PHGDH)

Docking studies of 169 phenolic compounds were performed to evaluate their affinities against both coenzyme and substrate binding sites of PHGDH. The binding cavities and their details for each molecular docking simulation using the MVD program package at the Nicotinamide adenine dinucleotide (NAD+) and 3-Phosphoglyceric acid binding sites are demonstrated in Figure 1 and Figure 2, respectively. Compounds already reported in the literature as inhibitors of PHGDH (BI-4924, NCT 503, and CBR-5884) [9,14,15] were used as control. The obtained docking scores of compounds with the Nicotinamide adenine dinucleotide (NAD+) and 3-Phosphoglyceric acid binding sites are presented in Table 1. A comparison of the binding energies of control compounds with test compounds clearly indicates that the hit compounds have better binding affinities with both binding sites of PHGDH than control compounds. The binding energy of Salvianolic acid C, Schizotenuin F, Vernolide B, and Salvianolic acid A at the NAD+ binding cavity scores were found to be −189.11 Å, −182.47 Å, −174.98 Å, and −173.90 Å, respectively. The MolDock scores of Salvianolic acid I, Chicoric acid, Salvianolic Acid C, and Calceolarioside D at the 3-Phosphoglyceric acid binding pocket were retrieved as −160.87 Å, −159.05 Å, −157.89 Å, and −150.78 Å, respectively. Table 2 shows the interaction details between four hit compounds (Salvianolic acid C, Schizotenuin F, Vernolide B, and Salvianolic acid A) and amino acid residues at the NADH binding site. The hit compound, Salvianolic acid C, binds with the human PHGDH in the NADH binding complex through conventional hydrogen bonds (ARG53, ARG74, THR77, ASN101, LEU208, ARG235, GLN291, ALA75, and ASP259), Pi-Alkyl (ALA234, ARG235, ALA285, PRO207, and ARG235), and Pi-Sigma (ILE155 and ALA234). Table 3 demonstrates the interaction detail between the four selected hit phenolic compounds (Salvianolic acid I, Chicoric acid, Scrophuloside B, and Salvianolic acid C) and amino acid residues at the substrate binding site of PHGDH. The top hit compound, Salvianolic acid I, interacted with the human PHGDH in the complex through conventional hydrogen bonds (ARG122, ARG170, ARG170, and VAL26), carbon–hydrogen bond (HIS62 and THR167), and Amide-Pi Stacked (GLY395 and GLY396).

#### 2.1.1. Computational Analysis of Physicochemical Properties of Hit Compounds

Compounds selection was based on the best docking score values and excellent binding interactions with PHGDH. The selected six hit compounds (Salvianolic acid I, Chicoric acid, Salvianolic acid C, Scrophuloside B, Schizotenuin F, and Vernolide B) were further evaluated for drug-likeness and physicochemical properties. The radar plot (Figure 3) clearly indicates that hit compounds have good physicochemical properties for oral applications. Other compounds meet these rules with few exceptions as shown in Table 4. For medicinal chemistry properties, these compounds comply with drug-likeness rules of the Pfizer criteria demonstrating that these compounds possess drug-like properties.

#### 2.1.2. ADMET Profiling of Hit Compounds

We further conducted ADMET profile (absorption, distribution, metabolism, excretion, and toxicity) studies for the six compounds with top docking scores against PHGDH using ADMETlab 2.0 online platform. ADMET properties of compounds are represented in Table 5. Except for Vernolide B, all other compounds possess low absorption profiles and good binding efficacy to blood plasma proteins. Chicoric acid, Salvianolic acid C, Scrophuloside B, and Schizotenuin F were predicted to be metabolized by CYP2C19. The prediction analysis demonstrated that these molecules have no effect on other human cytochrome P450 family enzymes, which indicates that these compounds could attain sustainable levels to inhibit PHGDH activity. It is also estimated that Chicoric acid and Scrophuloside B have low toxicity properties, further supporting the medicinal potential of these compounds. The maximum recommended daily doses (FDA MDD) for these compounds were also predicted as excellent. Furthermore, these compounds were also predicted as non-modulators of hERG indicating that these compounds will not affect cardiac activities.

### 2.2. Evaluation of Cytotoxic Potential of Hit Compounds against PHGDH Overexpressing Gastric Cancer Cells

Among hit compounds, a commercially available and assessable compound, Chicoric acid, was purchased to evaluate its anti-cancer potential against PHGDH overexpressing gastric cancer cell line MGC-803 and SGC-7901. The result of the MTT assay demonstrates that Chicoric acid was found to be cytotoxic against gastric cancer. Furthermore, the cytotoxicity of Chicoric acid was evaluated against PHGDH expressing cell lines (MGC-803 and SGC-7901) and PHGDH low expressing cell lines (MCF-7 and MDA-MB-231). The obtained results demonstrated that Chicoric acid possesses selective cytotoxicity toward MGC-803 and SGC-7901 cells as compared to MCF-7 and MDA-MB-231 (Figure 4).

## 3. Discussion

Targeting serine synthesis pathway enzymes has emerged as a novel opportunity for the selective inhibition of proliferation in cancer cells. PHGDH is the first rate-limiting enzyme that catalyzes the conversion of 3-PG into 3-PPyr in the first step of SSP [9]. PHGDH-mediated serine synthesis plays a critical role in metastasis and tumor growth. Cancer cells rely on both exogenous and endogenous serine for their proliferation. Various in-vivo studies demonstrated that deprivation of dietary serine has the potential to slow down the growth of cancer cells. According to the previously reported data on the xenograft model, serine- and glycine-free diet halted the progression of cancer growth in mouse models, indicating that dietary serine modifications might have the potential to cure cancer [16,17]. 

Thus, the metabolic inhibitors have the potential to target the PHGDH and halt the proliferation of cancerous cells. Thus, targeting PHGDH via metabolic inhibitors has the potential to halt the growth of cancerous cells [9]. 

In the current study, we have screened the natural products library against PHGDH by computational screening to identify metabolic inhibitors for PHGDH-amplified tumor types.

Among 169 virtually tested compounds, Salvianolic acid C, Schizotenuin F, and Vernolide B possess low binding energies for co-factor binding sites while the following compounds: Salvianolic acid I, Chicoric acid, Scrophuloside B, and Salvianolic acid C possess good binding potential to substrate binding site of PHGDH with acceptable ADMET profiles. Salvianolic acid C is a pharmacologically active polyphenolic compound derived from the highly valued traditional Chinese medicinal plant Salvia miltiorrhiza [18]. Our obtained results demonstrated that Salvianolic acid C interacts directly with the substrate as well as the co-factor binding site of PHGDH, thus, it has the potential to inhibit both receptor sites and act as a dual receptor inhibitor. Arginine was found to be a common interacting amino acid forming conventional hydrogen bonds and Pi-Alkyl interactions with salvianolic acid C. All the physiochemical parameters for salvianolic acid C were found to be within acceptable range except TPSA and LogP values. Salvianolic acid C can be structurally modified in such a way that these modifications will improve the physicochemical properties, however, its enzyme inhibitory activity will remain unaffected. Salvianolic acid C also possesses a comparatively good ADMET profile suggesting that it possesses drug-like properties. 

Chicoric acid is naturally found in aerial parts of various plants including Echinacea augustifolia and Echinacea purpurea [19]. Chicoric acid has been reported to possess anti-cancer activity against breast (MCF-7, SK-BR-3, MDA-MB-231), cervical (HeLa), colon (HT-29, HCT-116), and prostate (PC-3) cancers [20]. The obtained results for the anti-cancer activity of Chicoric acid are in line with the previous study reporting its anti-gastric cancer potential [21]. 

The knockdown of PHGDH in gastric cancer cells (SGC-7901 and MGC-803) leads to an elevated apoptosis percentage [22], suggesting that these gastric cancer cell lines are sensitive to PHGDH depletion as cancer cells with expression of PHGDH were found to be uniquely sensitive to PHGDH knockdown [9]. Messenger RNA and protein product PHGDH have been reported to exhibit gastric tissue preferred expression in samples from gastric cancer patients [23]. Comparison of our obtained IC_50_ values of Chicoric acid against PHGDH expressing cell lines MGC-803 and SGC-7901 (18 µM and 30 µM) and cell lines with low expression of PHGDH: MCF-7 and MDA-MB-231 (208 μM, 107 μM) clearly indicate that the Chicoric acid was found to be more potent against MGC-803 and SGC-7901 as compared to MCF-7 and MDA-MB-231. 

## 4. Materials and Methods 

### 4.1. Structure Retrieval and Preparation of PHGDH

3D X-ray diffraction (XRD) structures of human 3-Phosphoglycerate dehydrogenase were retrieved from the Protein Data Bank (PDB IDs 2G76 and 3C3C) along with NAD+ and 3-Phosphoglyceric acid. In the crystal structure, water molecules were removed and the structural error residues were checked rebuilt and optimized with the tools available in Molegro Virtual Docker (MVD). The prepared protein structure was saved as a receptor for docking analysis [24]. 

### 4.2. Phytochemical Library Preparation

For the preparation of the phytochemical library, the 3D SDF structures of compounds were taken up from the PubChem database (https://pubchem.ncbi.nlm.nih.gov/, accessed on 28 January 2022). BI-4924, NCT 503, and CBR-5884 were used as control compounds to compare results. The capacity of ligands to interact with the target sites of 3-Phosphoglycerate dehydrogenase was tested through the computational ligand-target docking method. Molegro Virtual Docker 6.0 on MolDock Score function was performed for molecular docking. Natural ligands in the crystal structure were set as the center of the docking region. In order to validate the docking studies, these compounds were re-docked inside the crystal structures of human PHGDH [25]. 

### 4.3. Docking Analysis

The capacity of ligands to interact with the target sites of 3-Phosphoglycerate dehydrogenase was tested through the computational ligand-target docking method. Molegro Virtual Docker 6.0 on MolDock Score function was performed for molecular docking. The conformation having low binding energy was evaluated as the most affirmative docking pose. The possible interactions between ligands and protein were checked using Discovery Studio Visualizer 2021 (Accelrys Software Inc., San Diego, CA, USA) [25]. The site finder tool available in MVD software was used to select the active site residues of the PHGDH bond (ARG122, ARG170, ARG170, and VAL26) [26]. After performing the docking, the compounds with the best binding affinity were selected for further evaluation. 

### 4.4. Cell Culture

The human gastric cancer cells (MG-803 and SGC-7901) and breast cancer cell line (MCF-7 and MDA-MB-231) were cultured in Dulbecco’s Modified Eagle’s Medium (DMEM) supplemented with 10% fetal bovine serum (FBS), penicillin and 100 μg/mL streptomycin and maintained at 37 °C with 5% CO2 in humidified atmosphere [27]. 

### 4.5. MTT Cytotoxicity Assay

Cell viability of the phenolic compounds was determined by MTT assay. Cancer cells were cultured in a 96-well plate. Briefly, cancer cells were treated with varying concentrations of drugs for 48 h. A 10µL MTT reagent was added (5 mg/mL) after 48 h and cells were further incubated at 37 °C for 4 h. Then 150 μL DMSO was added to dissolve formazan crystals and absorbance was measured at 570 nm in a microplate reader. [27]. 

The absorbance of the control cell and treated cells were calculated by the following equation.


I%=[A570(control)−A570(treated)]/A570(control)×100


### 4.6. ADMET and Drug Likeness Analysis of Compounds

ADMET prediction platforms allow us to predict some of the pharmacokinetic and drug-like properties related to compounds using structure similarity with the results from previous experimental studies. The compounds with best docking scores were further subjected to ADMET profiling. The compounds with physicochemical property hits were analyzed by using ADMETlab 2.0 (https://admetmesh.scbdd.com accessed on 28 January 2022) [28]. The canonical SMILES of the selected hits were exposed to ADMETlab [29].

## 5. Conclusions

In this study, computational molecular docking was performed using the phenolic compounds library against human PHGDH. Based upon the obtained in silico screening results, Salvianolic acid C was found to be the top ranking compound for the co-factor binding site and a common hit compound for both the substrate as well as the co-factor binding site of PHGDH. While Chicoric acid possesses good binding affinity to the substrate binding site of PHGDH. Chicoric acid was also found to possess in-vitro anti-cancer potential. Both compounds also have favorable ADMET properties. Thus, Salvianolic acid C and Chicoric acid could serve as promising lead compounds for the development of an anticancer agent with inhibitory activities against the human PHGDH. The development of such metabolic inhibitors could treat cancer patients with the amplification or overexpression of PHGDH. Although this study has unveiled the potential of Chicoric acid as a PHGDH modulator via virtual screening, biochemical and in-vitro studies are recommended to validate PHGDH as the direct molecular target of Chicoric acid. 

## Figures and Tables

**Figure 1 molecules-27-06108-f001:**
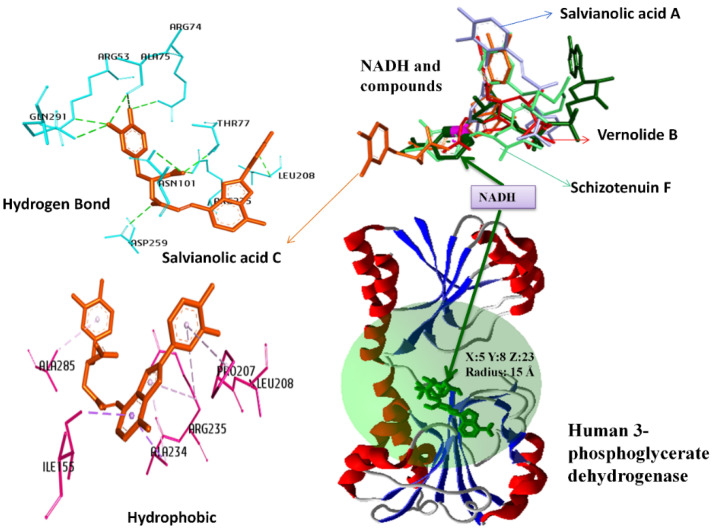
Binding cavities and their details for each molecular docking simulation (The nitrogen of the nicotinamide ring of NAD+ has been marked in pink and bold).

**Figure 2 molecules-27-06108-f002:**
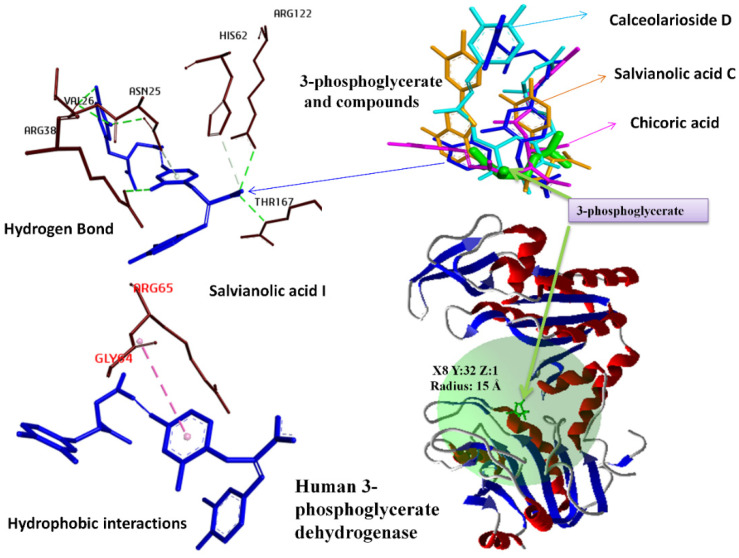
Binding cavities and their details for each molecular docking simulation.

**Figure 3 molecules-27-06108-f003:**
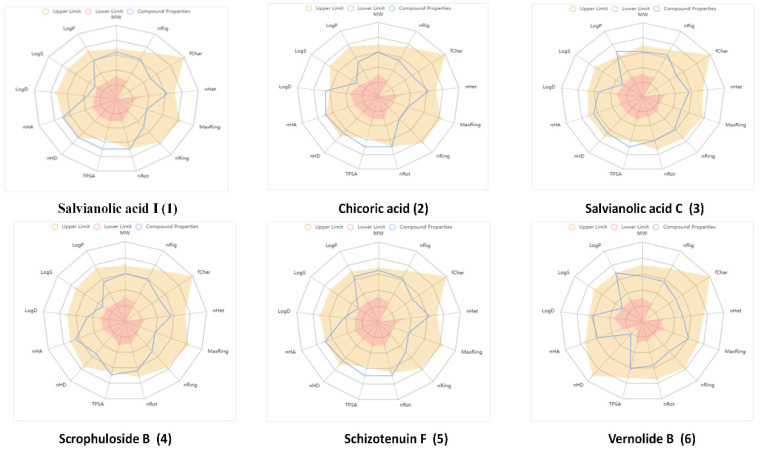
Radar plot demonstrating physicochemical properties of hit compounds. Brown Area: Upper Limit for each property, Blue area: Compound Property, Pink area: Lower limit of Physiochemical Property.

**Figure 4 molecules-27-06108-f004:**
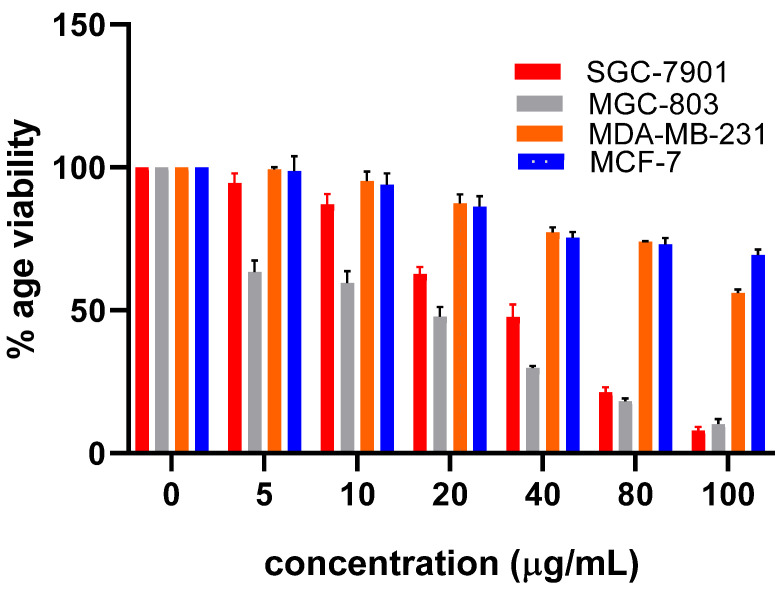
Dose-dependent growth inhibition of various cancer cell lines by Chicoric acid.

**Table 1 molecules-27-06108-t001:** Molecular docking score of hit phenolic compounds at the Nicotinamide adenine dinucleotide (NAD+) and 3-Phosphoglyceric acid binding sites of PHGDH.

Nicotinamide Adenine Dinucleotide (NAD^+^) Binding Cavity				3-Phosphoglyceric Acid Binding Cavity			
Compounds	PubChem CID	MolDock Score	HBond	Compounds	PubChem CID	MolDock Score	HBond
BI-4924	138756831	−154.91	−12.89	BI-4924	138756831	−122.20	−13.66
NCT 503	118796328	−144.58	−1.008	NCT 503	118796328	−117.15	−3.14
CBR-5884	4674993	−138.48	−4.401	CBR-5884	4674993	−113.39	−5.55
Salvianolic Acid C	13991590	−189.11	−19.18	Salvianolic acid I	10459878	−160.87	−14.28
Schizotenuin F	10347565	−182.47	−12.19	Chicoric acid	5281764	−159.05	−19.22
Vernolide B	73076547	−174.98	−4.77	Scrophuloside B	13991590	−157.89	−8.24
Salvianolic acid A	5281793	−173.90	−12.43	Salvianolic Acid C	14015431	−150.78	−19.58
Salvianolic Acid N	49769102	−168.28	−16.93	Tungtungmadic Acid	70697815	−146.01	−22.12
Salvianolic acid I	10459878	−168.17	−16.88	3,5-Di-O-caffeoyl-muco-quinic acid	6475855	−144.32	−19.87
Apigenin-7-O-gentiobioside	10841200	−167.10	−25.07	Salvianolicacid D	73060756	−141.63	−11.90
3,5-Di-O-caffeoyl-muco-quinic acid	6475855	−164.60	−13.56	Schizotenuin F	Unknown 1	−141.59	−18.25
Chicoric acid	5281764	−164.37	−19.62	Salvianolic acid A	5281793	−140.21	−13.02
Tungtungmadic Acid	70697815	−164.34	−14.49	Salvianolic acid K	10482829	−138.74	−17.69

**Table 2 molecules-27-06108-t002:** The type of interactions and amino acids involved in the binding at the NADH binding site.

Salvianolic Acid C (L1)				Schizotenuin F (L2)				Vernolide B (L3)				Salvianolic Acid A (L4)			
T	C	Interacting Group	D	T	C	Interacting Group	D	T	C	Interacting Group	D	T	C	Interacting Group	D
**Hydrogen Bond**	**Conventional Hydrogen Bond**	ARG53:NE—L1:O5	2.87	**Hydrogen Bond**	**Conventional Hydrogen Bond**	GLY78:N—L2:O11	3.10	**Hydrogen Bond**	**Co. HB**	THR77:OG1—L3:O8	3.10	**Hydrogen Bond**	**Conventional Hydrogen Bond**	ASP80:OD1—L4:O6	3.08
		ARG74:NH1—L1:O7	3.32			LEU208:N—L2:O8	2.99			GLY78:N—L3:O7	2.98			GLY153:N—L4:O4	2.91
		THR77:OG1—L1:O4	2.72			ARG235:NH1—L2:O4	2.76			L3:O6—CYS233:O	2.83			ARG154:NH1—L4:O2	3.08
		ASN101:ND2—L1:O4	3.27			ALA285:N—L2:O3	3.46		**Ca. HB**	ALA234:CA—L3:O6	3.30			GLY156:N—L4:O4	3.10
		LEU208:N—L1:O6	2.81			L2:O4—ALA234:O	2.78			L3:C14—PRO98:O	3.54			ARG235:NE—L4:O6	2.60
		ARG235:NH1—L1:O8	2.90			L2:O4—HIS282:NE2	3.07			L3:C16—ALA234:O	3.13			L4:O6—ASP80:OD2	3.27
		GLN291:NE2—L1:O5	3.16			L2:O12—THR206:O	2.97	**Hydrophobic**	**Alkyl**	ALA105—L3:C22	3.68			L4:O9—THR206:O	2.60
		L1:O5—ALA75:O	2.91	**Hydrophobic**	**Alkyl**	ALA105—L2	4.08			ARG154—L3	4.61			L4:O10—CYS233:O	3.37
		L1:O7—ALA75:O	3.22			ILE155—L2	4.99			PRO207—L3	5.32		**Ca. HB**	ARG154:CD—L4:O2	3.06
		L1:O10—ASP259:OD2	3.39			PRO207—L2	5.00			L3—ILE155	4.52		**Pi-DHB**	ARG235:NH1—L4	4.02
**Hydrophobic**	**Pi-Alkyl**	L1—ALA234	4.55			ARG235—L2	5.46			L3:C9—ARG154	3.92	**Hydrophobic**	**Pi-Alkyl**	L4—ILE155	5.21
		L1—ARG235	5.08							L3:C9—ILE155	4.82			L4—ALA234	4.57
		L1—ALA285	4.09							L3:C10—PRO207	4.48			L4—ARG235	5.05
		L1—PRO207	4.74							L3:C19—PRO207	5.03				
		L1—ARG235	5.35							L3:C19—ARG235	2.94				
	**Pi-S**	ILE155:CD1—L1	3.70							L3:C23—LEU208	4.91				
		ALA234:CB—L1	3.81							L3:C23—ARG235	4.11				

T: Type C: Category, D: Distance (Å), H:Hydrophobic, Co. HB: Conventional Hydrogen Bond, Ca. HB: Carbon Hydrogen Bond, Pi-S: Pi-Sigma.

**Table 3 molecules-27-06108-t003:** Type of interactions and amino acids involved in the binding at the 3-Phosphoglycerate site.

Salvianolic Acid I (L5)				Chicoric Acid (L6)				Scrophuloside B (L1)				Salvianolic Acid C (L7)			
T	C	Interacting Group	D	T	C	Interacting Group	D	T	C	Interacting Group	D	T	C	Interacting Group	D
**Hydrogen Bond**	**Conventional Hydrogen Bond**	ARG122:HH12—L5O11	2.08	**Hydrogen Bond**	**Conventional Hydrogen Bond**	ASN25:HD22—L6:O8	1.81	**Hydrogen Bond**	**Conventional Hydrogen Bond**	ARG65:HE—L1:O4	2.80	**Hydrogen Bond**	**Conventional Hydrogen Bond**	ASN25:HD21—L7:O1	2.089
		ARG170:HE—L5O11	1.66			ARG38:HH12—L6:O8	2.72			ARG65:HH21—L1:O4	1.88			ARG38:HH22—L7:O4	2.311
		ARG170:HH21—L5O11	2.76			ARG65:HN—L6:O9	1.73			ARG122:HH12—L1:O8	1.69			ARG65:HN—L7:O9	2.258
		L5H1—VAL26:O	1.70			ARG65:HE—L6:O5	2.28			ARG170:HE—L1:O2	2.24			ARG122:HH22—L7:O2	2.139
		L5H2—VAL26:O	1.74			ARG65:HH21—L6:O3	1.44			ARG170:HH21—L1:O4	2.20			ARG170:HH21—L7:O5	2.768
	**Ca. HB**	HIS62:HE1—L5O12	2.95			ARG122:HH12—L6:O5	2.55			L1:H6—ASP23:OD2	2.97			L7:H1—GLY166:O	2.072
		THR167:HA—L5O12	2.56			ARG122:HH12—L6:O6	1.63			L1:H6—GLY166:O	2.09			L7:H2—GLY166:O	1.651
**H**	**A-Pi S**	GLY64:C,O; ARG65:N—L5	5.21			ARG122:HH22—L6:O5	1.58		**Ca. HB**	HIS62:HE1—L1:O10	2.35			L7:H5—ASN25:O	2.562
						ARG170:HE—L6:O4	1.57			GLY64:HA2—L1:O7	2.23			L7:H6—TYR75:OH	2.991
						ARG170:HH21—L6:O3	1.76			HIS172:HE1—L1:O8	2.77			L7:H7—MET28:O	2.472
						ARG170:HH21—L6:O4	2.68	**HB-E**	**Pi-C**	ARG38:NH1—L1	3.32		**Ca. HB**	HIS62:HE1—L7:O3	3.065
						L6:H2—ASP23:OD2	2.08		**Pi-D HB**	ARG38:NH2—L1	3.46			GLY64:HA1—L7:O9	3.087
						L6:H3—TYR75:OH	2.29	**E**	**Pi-C**	ARG65:NH2—L1	3.99			L7:H11—HIS62:NE2	2.184
						L6:H4—GLY372:O	3.06	**HB**	**Pi-DHB**	ASN25:HD21—L1	2.64	**H**	**Pi-A**	L7—LEU63	5.042
					**Ca. HB**	HIS169:HE1—L6:O4	2.90	**Hydrophobic**	**Pi-A**	L1—ARG65	4.97				
						L6:H8—HIS62:NE2	2.09		**Pi-A**	L1—PRO27	4.74				
				**H**	**A-Pi S**	GLY395:C,O; GLY396:N—L6	4.02								
					**Pi-A**	L6—ARG65	3.90								

T: Type C: Category, D: Distance (Å), H:Hydrophobic, HB-E: Hydrogen Bond; Electrostatic, E: Electrostatic, Ca. HB: Carbon Hydrogen Bond, Pi-Pi S: Pi-Pi Stacked, A-Pi S:Amide-Pi Stacked, Pi-A: Pi-Alkyl, Pi-DHB: Pi-Donor Hydrogen Bond, Pi-C: Pi-Cation, Pi-C: Pi-D HB: Pi-Cation; Pi-Donor Hydrogen Bond.

**Table 4 molecules-27-06108-t004:** Table of physicochemical properties of six hit compounds.

Physicochemical Property	Optimal	1	2	3	4	5	6
Molecular Weight (MW)	100~600	538.11	474.08	492.11	474.15	552.13	434.19
nHA	0~12	12	12	10	10	12	8
nHD	0~7	7	6	6	4	6	1
nRot	0~11	11	11	8	9	12	7
nRing	0~6	3	2	4	3	3	3
MaxRing	0~18	6	6	9	6	6	13
nHet	1~15	12	12	10	10	12	8
fChar	−4~4	0	0	0	0	0	0
nRig	0~30	23	18	25	21	23	19
TPSA	0~140	**211.28**	**208.12**	**177.89**	**151.98**	**200.28**	104.43
logS	−4~0.5	−3.679	−3.503	−3.867	−3.406	−3.875	**−4.02**
logP	0~3	2.074	1.672	**3.641**	1.846	2.491	2.973
logD	1~3	1.389	**3.427**	2.265	1.808	1.643	2.774
Medicinal chemistry	Pfizer Rule	Accepted	Accepted	Accepted	Accepted	Accepted	Accepted

Number of Hydrogen Bond Acceptors, (nHA), Number of Hydrogen Bond Donors (nHD), Total Polar Solvent Accessibility (TPSA), water solubility (log S), Octane/water partition coefficient (LogP), distribution coefficient (LogD). (Bold: indicates those exceeding optimal conditions).

**Table 5 molecules-27-06108-t005:** The ADMET profile of hit compounds.

Category	Property	1	2	3	4	5	6
Absorption	Caco-2 > −5.15	−6.271/poor	−6.450/poor	−5.629/poor	−5.966/poor	−5.954/poor	−4.772/excellent
	Pgp-Inhibitor	---	---	---	---	---	--
	Pgp-Substrate	---	---	---	--	---	---
	HIA	+	+	+	++	+	---
Distribution	PPB	96.077%	98.828%	99.640%	98.413%	96.746%	86.919%
	BBB	---	---	---	--	---	-
Metabolism	CYP1A2-Inhibitor	-	---	+	-	+	--
	CYP1A2-Substrate	---	---	---	---	---	+++
	CYP3A4-Inhibitor	---	---	--	-	--	--
	CYP3A4-Substrate	---	---	---	--	---	+
	CYP2C19-Inhibitor	+	-	+	+	+	+
	CYP2C19-Substrate	--	+++	++	++	++	--
	CYP2C9-Inhibitor	---	---	---	--	---	---
	CYP2D6-Inhibitor	--	--	--	+	--	--
	CYP2C9-Substrate	---	---	--	--	--	+
	CYP2D6-Substrate	---	--	---	--	---	+
Excretion	Clearance	3.954/Low	4.971/Low	10.679/ Moderate	5.328/ Moderate	8.282/ Moderate	11.696/ Moderate
Toxicity	hERG	---	---	---	-	---	---
	H-HT	++	--	--	---	-	++
	Ames	---	---	--	--	---	+++
	DILI	++	--	+++	--	+++	+++
	FDAMDD	---	---	--	---	--	---
For the classification models, the prediction probability values are represented with different symbols: 0–0.1(---), 0.1–0.3(--), 0.3–0.5(-), 0.5–0.7(+), 0.7–0.9(++), and 0.9–1.0(+++). Usually, the token ‘+++’ or ‘++’ represents the molecule that is more likely to be toxic or defective, while ‘--’ or ‘-’ represents nontoxic or appropriate.							

**Absorption:** Caco-2: Caco-2 cell permeability; P-gp: the estimation of compound being substrate and inhibitor of P-glycoprotein; HIA: Human intestinal absorption. Distribution: PPB: plasma protein binding; BBB: the prediction of compound to cross blood-brain barrier. Metabolism: The estimation of compound being inhibitor or substrate of cytochrome P450 isozymes. Excretion: Clearance from the body. H-HT: Human hepatotoxicity; AMES, Compound having mutagenic activity; FDA MDD: Maximum FDA recommended daily doses.

## Data Availability

Authors may be contacted for more details concerning data supporting reported results.

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
