# Peer review of "Identification of Novel Natural Inhibitors to Human 3-Phosphoglycerate Dehydrogenase (PHGDH) for Cancer Treatment"

_molecules, 2022, doi:10.3390/molecules27186108_

Round 1

Reviewer 1 Report

The manuscript titled ‘Identification of Novel Natural Inhibitors to Human 2 3-phosphoglycerate Dehydrogenase (PHGDH) for 3 Cancer Treatment’, tested virtually 169 natural compounds and found some molecules with interesting potential to bind to the co-factor binding site of PHGDH, a key enzyme that catalyzes the conversion of 3-PG into 3-PPyr in the first step of serine synthesis pathway, playing therefore, a critical role in cancer progression. However, the main concern of the study is to confirm the ability of these molecules to bind to their target, below are some comments related to this:

Even with a cytotoxic effect on gastric cancer cell line MGC-803, there is no evidence that the effect of chicoric acid is related to its capacity to bind and block PHGDH

Comparing the IC50 obtained in your study with the IC50 on MCF-7 cell line of another study could lead to biased conclusions. The comparison should be conducted in the same study and same conditions

At least, 3 molecules with different binding efficacy should be compared using the gastric cancer cell line MGC-803 and a PHGDH-independent cell lines. These comparisons could help you support the hypothesis that the effect observed on MGC-803 cells is due to the interaction with PHGDH.

You should also assess the activity of PHGDH in vitro with and without the inhibitors.

Author Response

Respected Reviewer,

Reviewer 2 Report

This paper describes the use of in-silico strategies to determine the potential PHGDH inhibitors by screening several natural polyphenolic compounds. This topic is of particular interest to the broader tumor metabolism community. However, this work need considerable amount of work in order to publish in Journal Biomolecules. Some of the comments are given point by point-

1) PHDGH is considered as a rate-limiting step in serine synthesis pathway and several labs has shown that in different cancer types. However, authors have failed to cite some of the relevant paper showing the important of PHDGH as rate limiting step in SSP. Please cite relevant literature such as  PMID: 27110680 in addition to several other.

2) Authors should discuss how and why they think the inhibitors discovered in this paper would be better especially when several PHGDH inhibitors are already developed by different labs. One example is NCT-503.

3) Docking score for both of the cavities for NAD+ and 3-PG is really low suggesting that the administered drug can bind to both of them strongly upon drug administration. Author should show the inhibition of PHGDH inhibition by performing in vivo and in vitro experiments. This can be done by using either GC-MS or using some calorimetric bases assay. 

4) It has been suggested that vernolide b posseses great oral application, What is the criteria for selecting only vernolide b as the best candidate while all other also passe tha standards?

5) If vernolide is the best candidate, please determine the IC50 for vernolide B also.

6) Please indicate the red area in adar plot.

7) Authors used a gastric cancer cell line to test Chicoric acid. However, it is not clear they chose this cell line. Please provide corresponding publicly TCGA data for PHGDH expression in gastric cancer patients. Also, please perform these experiment more than one cell lines and compare the expression by WB and RT to normal gastric cells. Author also mentioned about other cell SGC-7901 in material and methods. But never showed the data related to that.

8) Authors should also discuss about the pathway redudancy in serine synthesis pathway more clearly and possibly talk about the dietary modifications as possible therapy for cancers.

9) In addition , author are discusing that comparison of our obtained IC50 values 188 of chicoric acid MGC803 (18 μM) with PHGDH-independent cell lines: MCF-7,  MDAMB231 (208 μM, 107 μM).  This statement of MCF7- cell is not true. MCF7 cells are not sensitive to PHGDH inhibtion because they do not expression PSAT1 enzyme ( another important enzyme in serine synthesis pathway). Author can have a look at this paper- PMID: 35045283.

Author Response

Respected Reviewer,

Reviewer 3 Report

Nice study and very good presentation of the hypothesis. Chicoric acid appears to be a better fit at the 3-phosphoglyceric acid than the Nicotinamide adenine dinucleotide.

·         Page 2, line 78: are these high or extra precision scores, please mention the iterations. As the different softwares followed have different unit outputs. So please add the unit of the scores and iterations, etc.

·         General query, is there a control compound docked for both the targets, may be this can be added in supporting info. It can be something known drug with known binding affinity or something which does not bind at all.

·         Figure 1, and figure 2 high resolution images are needed. The interaction distance in Angstrom unit is preferred way.

·         Chicoric acid is also know against gastric cancers, please add the references. Biomedicine & Pharmacotherapy 118 (2019) 109144.

Author Response

Respected Reviewer,

Round 2

Reviewer 1 Report

The authors have made extensive changes and conducted new experiments to validate the results obtained. There are still some minor issues to be fixed:

The authors need to re-read the whole document to check some typos, for exemple; line 211: Chicoric acid 'hs' been reported to possess.

The authors need to indicate the difference between the breast cancer subtypes as in the paper: https://doi.org/10.3390/app12041866

'The luminal subtype line (MCF-7) and the triple-negative breast cancer (TNBC) subtype line (MDA-MB2-31)'

Reviewer 2 Report

The paper is ok to accept now.

Reviewer 3 Report

the quries are well addressed.